# Fatty acyl-CoA reductase influences wax biosynthesis in the cotton mealybug, *Phenacoccus solenopsis* Tinsley

Haojie Tong [1,4], Yuan Wang[1,4], Shuping Wang[2], Mohamed A. A. Omar [1,3], Zicheng Li[1], Zihao Li[1], Simin Ding[1], Yan Ao[1], Ying Wang[1], Fei Li [1✉] & Mingxing Jiang [1✉]

Mealybugs are highly aggressive to a diversity of plants. The waxy layer covering the outermost part of the integument is an important protective defense of these pests. However, the molecular mechanisms underlying wax biosynthesis in mealybugs remain largely unknown. Here, we analyzed multi-omics data on wax biosynthesis by the cotton mealybug, *Phenacoccus solenopsis* Tinsley, and found that a fatty acyl-CoA reductase (*PsFAR*) gene, which was highly expressed in the fat bodies of female mealybugs, contributed to wax biosynthesis by regulating the production of the dominant chemical components of wax, cuticular hydrocarbons (CHCs). RNA interference (RNAi) against *PsFAR* by dsRNA microinjection and allowing mealybugs to feed on transgenic tobacco expressing target dsRNA resulted in a reduction of CHC contents in the waxy layer, and an increase in mealybug mortality under desiccation and deltamethrin treatments. In conclusion, *PsFAR* plays crucial roles in the wax biosynthesis of mealybugs, thereby contributing to their adaptation to water loss and insecticide stress.

[1] Ministry of Agriculture Key Lab of Molecular Biology of Crop Pathogens and Insect Pests, Institute of Insect Sciences, College of Agriculture and Biotechnology, Zhejiang University, Hangzhou, China. [2] Technical Centre for Animal, Plant, and Food Inspection and Quarantine, Shanghai Customs, Shanghai, China. [3] Department of Plant Protection, Faculty of Agriculture (Saba Basha), Alexandria University, Alexandria 21531, Egypt. [4] These authors contributed equally: Haojie Tong, Yuan Wang. ✉email: lifei18@zju.edu.cn; mxjiang@zju.edu.cn

Mealybugs (Hemiptera: Coccoidea: Pseudococcidae) are globally distributed, highly aggressive pests of many commercial, non-commercial, and ornamental plants[1–3]. They are a diverse group of sap-sucking insects, with at least 32 genera and around 105 species[4]. Mealybugs can attack over 300 plant species, with host records extending to 76 families and over 300 genera[5–7]. Both adults and crawlers suck the cell sap from the phloem of leaves, stems, twigs, flower buds, and young bolls, causing withering and yellowing of leaves, dieback, and even death of the whole plant. Over the years, mealybugs have caused significant economic loss worldwide, with reported crop losses reaching 52% in India resulting from infestation by *Formicoccus polysperes* Williams[8], and in Kenya, papaya yield losses were estimated at 57% with an annual economic loss of US $3,009 per ha due to the papaya mealybug *Paracoccus marginatus* Williams & Granara de Willink[9].

Unlike adult males that have an elongated body with wings but no wax, female mealybugs are globose, flattened, wingless, and typically covered by a layer of thick, powdery wax. This waxy layer is comprised of a complex mixture of lipids including cuticular hydrocarbons (CHCs), fatty acids, esters, alcohols, and ketones[10]. Among these chemicals, the CHCs may vary tremendously in their composition between different mealybug species[11], which implies that CHCs are closely correlated to the biological functions of the waxy layer. Considering the vital protective roles that the waxy layer provides against water loss and exposure to toxic substances in the environment[12,13], destruction of this waxy layer, through methods such as applications of wax-degrading bacteria, has been considered a feasible means to manage mealybugs[14,15]. A more thorough understanding of mealybug wax biosynthesis could illuminate novel targets and means of destroying the waxy layer, however, the underlying molecular mechanisms of this biosynthetic process remain largely unknown.

CHCs are one of the most important classes of cuticular lipids in insects, and their biosynthetic pathways have been well-studied since the early 1990s[16,17]. The pathways can be briefly divided into four steps: formation of fatty acid precursors, elongation into long-chain fatty acyl-CoAs, conversion to alcohols, and oxidation to aldehydes, which are respectively catalyzed by fatty acid synthase (FAS), fatty acid elongases (ELO), fatty acyl-CoA reductase (FAR), and cytochromes P450[18]. Among these enzymes, FARs have been identified in several insect species but they exhibit varying biological functions. In *Drosophila*, one *FAR* gene— *waterproof* (*DmWP*)—plays a key role in long-chain fatty acid metabolism that ultimately affects the formation of the outermost tracheal cuticle sublayer, termed the envelope, in embryos[19]. A total of 17 *FAR* genes (*NIFARs*) were identified in *Nilaparvata lugens* Stål by Li et al.[20,21] by injection with targeted double-stranded RNAs (dsRNA): four had crucial roles in cuticle shedding, two were involved in the production of the CHCs required for waterproofing, and eight were essential for female fertility. Additionally, the *AmFAR1* gene in *Apis mellifera* L.[22] and *FARs* in some lepidopterans[23–26] was reported to primarily affect pheromone biosynthesis by regulating the output of acetate or aldehyde derivatives. In scale insects (Hemiptera: Coccoidea), Hu et al.[27] found the *EpFAR* gene of *Ericeru pela* Chavannes to be highly expressed in the cuticle and functioned by converting acyl CoA to its corresponding alcohol. In the cotton mealybug, *Phenacoccus solenopsis*, Li et al.[28] identified two *FARs* (*PsFARs*), of which only one (*PsFAR* I) experienced increased relative expression levels after spirotetramat treatment; however, the functions of *FARs* in this mealybug remain undetermined due to a lack of direct functional assays.

*P. solenopsis* is an invasive pest in Asia and beyond[29], possessing almost all of the features unique to mealybugs (Fig. 1a). It causes serious economic losses to over 150 crop and horticultural plant species and can be especially detrimental to cotton production[7,30,31]. It is in this mealybug species that we chose to study the molecular mechanisms of wax biosynthesis. We first obtained the wax metabolome, integumentary transcriptome, and proteome data of cotton mealybugs. Using combined analyses of these omics data, a new *FAR* gene, *PsFAR*, was successfully identified. This gene was highly expressed in third-instar nymphs and adult females. Knocking down *PsFAR* with both microinjection and transgenic tobacco-mediated RNA interference (RNAi), we observed reduced wax production associated with changes in CHC synthesis, which lead to lower water retention and higher mortality from insecticide treatments in the cotton mealybug.

## Results

**Hydrocarbons are the dominant chemical components in wax.** The chemical components of mealybug cuticular wax vary by species[11] and are speculated to influence the biological functions of the wax, thus it is of great importance to obtain a complete catalog of chemical components in the mealybug wax. Here, we used *n*-hexane and methanol independently to dissolve the powdery wax collected from the body surface of female cotton mealybugs, then tested it on polar and nonpolar chromatographic columns using GC–MS. At least eight groups of chemical components were identified: hydrocarbons, fatty acids, olefins, alcohols, aromatic derivatives, esters, ketones, and aldehydes. Their relative contents varied between the four dissolvent-column treatments (Fig. 1b and Supplementary Fig. 1), where the top three components of each of the four treatments were (1) hydrocarbons, fatty acids, and olefins (*n*-hexane, polar column), (2) hydrocarbons, alcohols and fatty acids (methanol, polar column), (3) hydrocarbons, aromatic derivatives and esters (*n*-hexane, nonpolar column), and (4) hydrocarbons, alcohols and aromatic derivatives (methanol, nonpolar column). Among them, hydrocarbons (CHCs) made up 59–68% of the chemical profiles of the four treatments and were mainly composed of $C_{13}$–$C_{36}$ *n*-alkanes and $C_{12}$–$C_{20}$ branched chemicals (Supplementary Data 1). These results revealed that CHCs are the dominant components of cotton mealybug wax.

**Screening of the *PsFAR* gene in the CHC biosynthesis pathway.** To identify genes playing crucial roles in CHC biosynthesis in cotton mealybugs, we first performed transcriptome sequencing of integument tissues, which contain adherent waxy glands and fat bodies, and the remaining tissues (without integument) of adult female cotton mealybugs (Supplementary Table 1 and Supplementary Data 2). In the integument, 527 upregulated and 34 downregulated differentially expressed genes (DEGs) were found (Supplementary Fig. 2 and Supplementary Data 2). Among the significantly upregulated DEGs, we focused on those that were annotated as the three key CHC synthesis pathway-related enzymes, i.e., FAS, ELO, and FAR. Six such DEGs in the integument were identified, including two *FAS* genes (gene i.d.: PSOL09676 and PSOL12170), two *ELO* genes (gene id: PSOL06704 and PSOL00600), and two *FAR* genes (gene id: PSOL02039 and PSOL06156) (Supplementary Table 2, Supplementary Data 2).

We next performed proteome sequencing of the integument from adult female mealybugs. When the six DEGs described above were mapped to the proteomics data, at least one protein sequence for each could be BLASTed with an *e*-value < 2E−92 (Supplementary Table 3), confirming the expression of these genes in the integument. In terms of wax biosynthesis, only one *FAR* gene (gene id: PSOL02039) was found expressed at a

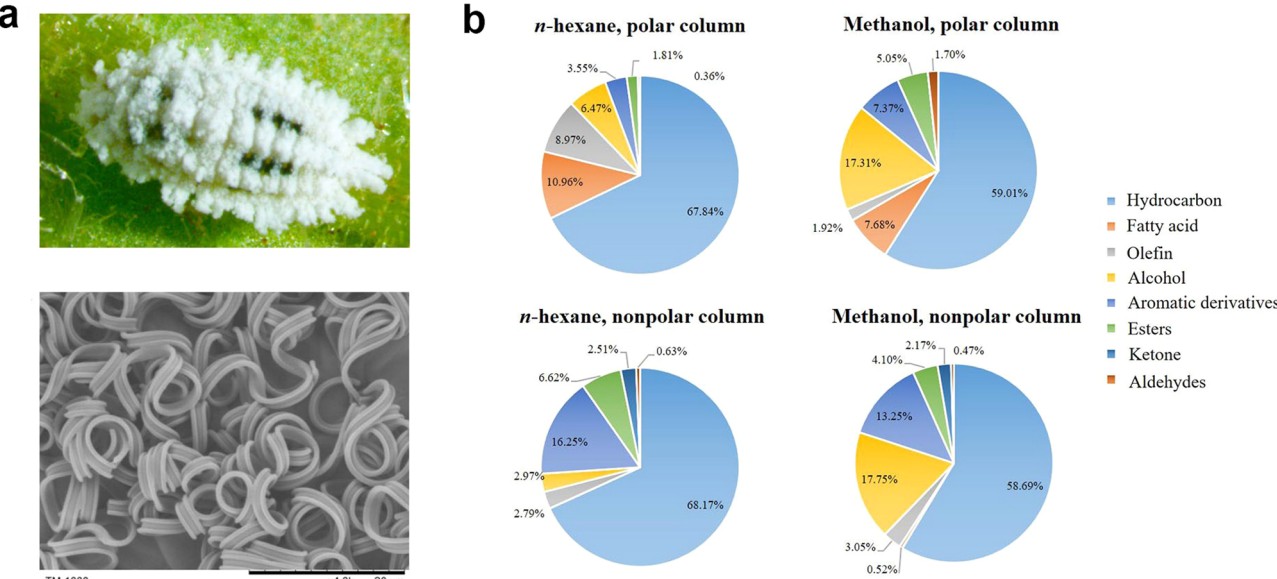

**Fig. 1 Mealybug wax and its relative chemical composition in adult *P. solenopsis* females. a** The wax covering on female *P. solenopsis* (top) appears filamentous under SEM (×4000, below). **b** Relative contents of identified chemical components based on GC–MS analysis. Wax, respectively, dissolved in *n*-hexane and methanol were analyzed on polar and nonpolar chromatographic columns, producing four test results. The solvent and column type are shown on top of each pie chart.

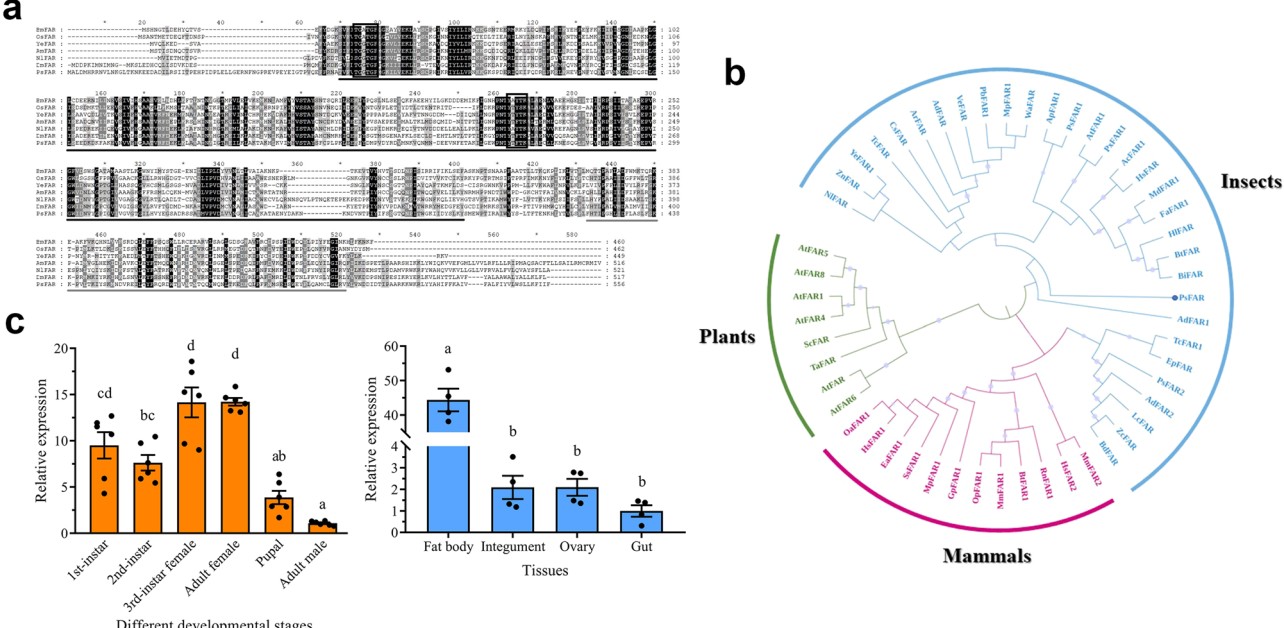

**Fig. 2 Sequence analysis and expression profiles of *PsFAR*. a** Multiple alignments of *FAR*s from *P. solenopsis* and other organisms. Bm: *Bombyx mori*, Os: *Ostrinia scapulalis*, Ye: *Yponomeuta evonymellus*, Am: *Apis mellifera*, Nl: *Nilaparvata lugens*, Dm: *Drosophila melanogaster*. Identical residues are shaded black, and conservative substitutions are shaded gray. The FAR contains a characteristic N-terminal Rossmann fold NAD(P) (+)-binding domain (black underline), a sterile alpha motif protein domain (gray underline), a cofactor binding motif (TGXXGF), and an active site motif (YXXXK) (black boxes). **b** Phylogenetic tree of FARs from different species. The tree was constructed using amino acid sequences with the neighbor-joining algorithm in MEGA. Differently colored sectors of the circles represent three main branches of insects (blue), mammals (red), and plants (green). The protein sequence of PsFAR, marked by a blue circle, clusters with insect FARs. Branches marked with gray circles indicate a confidence value larger than 0.50. Detailed information on the species used here can be found in Supplementary Data 3. **c** Developmental (left) and tissue-specific (right) expression profiles of *PsFAR* in *P. solenopsis*. The relative expression level was normalized and visualized as mean ± SEM with four biological replicates. Different lower-case letters indicate a significant difference ($P < 0.05$) based on ANOVA followed by Tukey's multiple comparison test.

significantly high level in female nymphs and adults (Fig. 2 and Supplementary Fig. 3). These results suggest that this *FAR* gene (named *PsFAR*) has crucial functions in CHC biosynthesis in the cotton mealybug.

**Sequence analysis and expression profiles of *PsFAR*.** Prior to verifying gene functions, the full-length sequence of *PsFAR* was obtained by rapid amplification of cDNA ends (RACE). This gene is composed of a 1671 bp ORF located downstream of a 177-bp 5'

UTR and upstream of a 78-bp 3' UTR (GenBank accession number: OM371326). The ORF encodes 556 amino acid residues with a predicted molecular mass of 64.2 kD and an isoelectric point of 8.6. Multiple sequence alignment showed that FAR sequences shared a characteristic N-terminal Rossmann fold NAD(P) (+)-binding domain, a sterile alpha motif protein domain, a cofactor binding motif (TGXXGF), and an active site motif (YXXXK) (Fig. 2a). To investigate the relationship of various *PsFAR* genes, we constructed a phylogenetic tree for *FAR*s from 40 other species (Fig. 2b). Three main branches, representing plant, mammal, and insect, *FAR*s were clustered in the tree, and *PsFAR* clustered with other insects *FAR*s as expected.

We determined the expression profiles of *PsFAR* at various developmental stages (first, second, and third-instar nymphs, pupa, adult males, and adult females) and tissues (integument, fat body, gut, and ovary) (Fig. 2c). Expression of this gene was highest in third-instar nymphs and adult females and lowest in adult males. It was significantly highly expressed in the fat body relative to the other three tissues in adult females, indicating that *PsFAR* plays important functions in the fat body of female cotton mealybugs.

**RNAi knockdown of *PsFAR* by dsRNA microinjection and feeding on transgenic tobacco.** We performed RNAi experiments to knockdown *PsFAR* expression using two strategies, conventional microinjection of *PsFAR* complementary dsRNA and feeding on transgenic tobacco expressing dsRNA. The ds*PsFAR* expression cassette pCAMBIA1301-ds*PsFAR* vector is shown in Fig. 3a. A total of 26 tobacco transformants (generation $T_0$) were produced after *Agrobacterium*-mediated transformation, incubation, differentiation, and rooting (Fig. 3b). To identify positive transformants among the 26 $T_0$ tobacco plants, a 557-bp fragment of the selective marker gene *hygromycin B phosphotransferase* (*Hpt*) were amplified using PCR, which suggested a

transformation efficiency of 100% (Fig. 3c and Supplementary Fig. 4). To screen tobacco transformants with the highest expression of ds*PsFAR*, we determined the relative expression levels in nine randomly selected $T_0$ genetically modified (GM) tobacco plants using RT-qPCR. Two transformants, #19 and #26, showed the highest ds*PsFAR* expression levels (Fig. 3d); the #19 GM tobacco line was selected for further analysis.

To determine the RNAi efficiency of dsRNA microinjection and GM tobacco feeding, we assessed the resulting *PsFAR* gene expression in cotton mealybugs under different treatments by RT-qPCR. The results showed that *PsFAR* expression was successfully suppressed 3 days after dsRNA microinjection (reduced to $54.8 \pm 3.7\%$) and 5 days after feeding on GM tobacco (reduced to $43.0 \pm 1.1\%$) (Fig. 4a). Meanwhile, no significant changes were detected in the expression levels of other *PsFAR* genes, including *PsFAR* (ID: PSOL06156), *PsFAR* I, and *PsFAR* II (Supplementary Fig. 5). Mortality also occurred in both RNAi treatments, with insects exhibiting abnormal coloration, reduced wax secretion, and unsuccessful molting after injection with ds*PsFAR* or feeding on ds*PsFAR* GM tobacco (Fig. 4b). Moreover, significant ($P < 0.05$) declines in survival rates were observed from day 2 (9.6%) to day 14 (20.2%) after dsRNA injection, and from day 5 (11.1%) to day 14 (25.6%) after feeding on GM tobacco (Fig. 4c), demonstrating some degree of lethality in cotton mealybugs after knocking down *PsFAR* expression.

***PsFAR* is required for CHC biosynthesis and wax output.** To explore whether *PsFAR* is responsible for CHC biosynthesis in the cotton mealybug, we used GC–MS to determine the composition and contents of CHCs in mealybug wax 4 days after dsRNA injection and 9 days after feeding on GM tobacco. The results showed that the wax contained a mixture of several *n*-alkanes with different chain lengths (Fig. 5a). The same types and

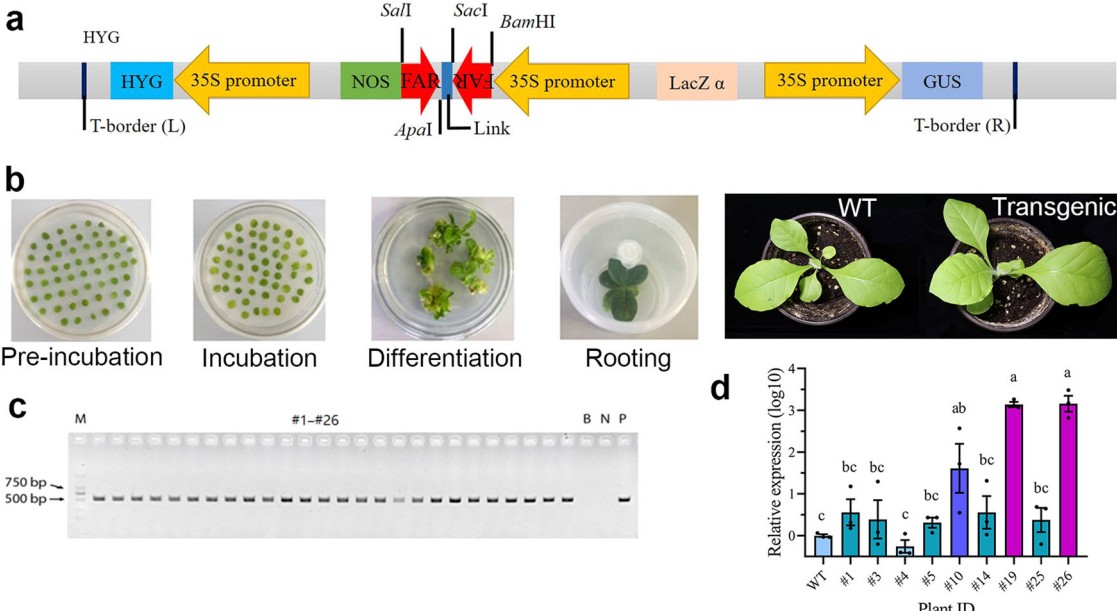

**Fig. 3 Genetically modified tobacco and ds*PsFAR* expression in transformants. a** Schematic diagram of the ds*PsFAR* expression cassette in the pCAMBIA1301-ds*PsFAR* vector introduced into *Nicotiana tabacum* cv. Petit Havana via *Agrobacterium*-mediated transformation. Two arrows in the opposite direction, respectively, indicate sense and antisense strands of ds*PsFAR*. HYG: hygromycin; NOS: Nos terminator; GUS: β-glucuronidase. **b** Tobacco leaf discs were pre-incubated in medium with kanamycin and then transferred to medium for differentiation and rooting. Seedlings were transferred into nutrient soil and grown in a greenhouse to obtain the $T_1$ generation. WT: wild-type plants, Transgenic: transgenic plants. **c** Selection of positive tobacco transformants via PCR amplification of the *hpt* gene. The first lane is marker (M), followed by 26 transformants (#1–#26), blank control (B), negative control (N), and Positive control (P). **d** Relative expression level of ds*PsFAR* in nine randomly selected $T_0$ GM tobacco. Data were normalized and are represented as the mean ± SEM from three biological replicates. The different letters indicate significant differences determined by one-way ANOVA followed by the Tukey's multiple comparison test ($P < 0.05$).

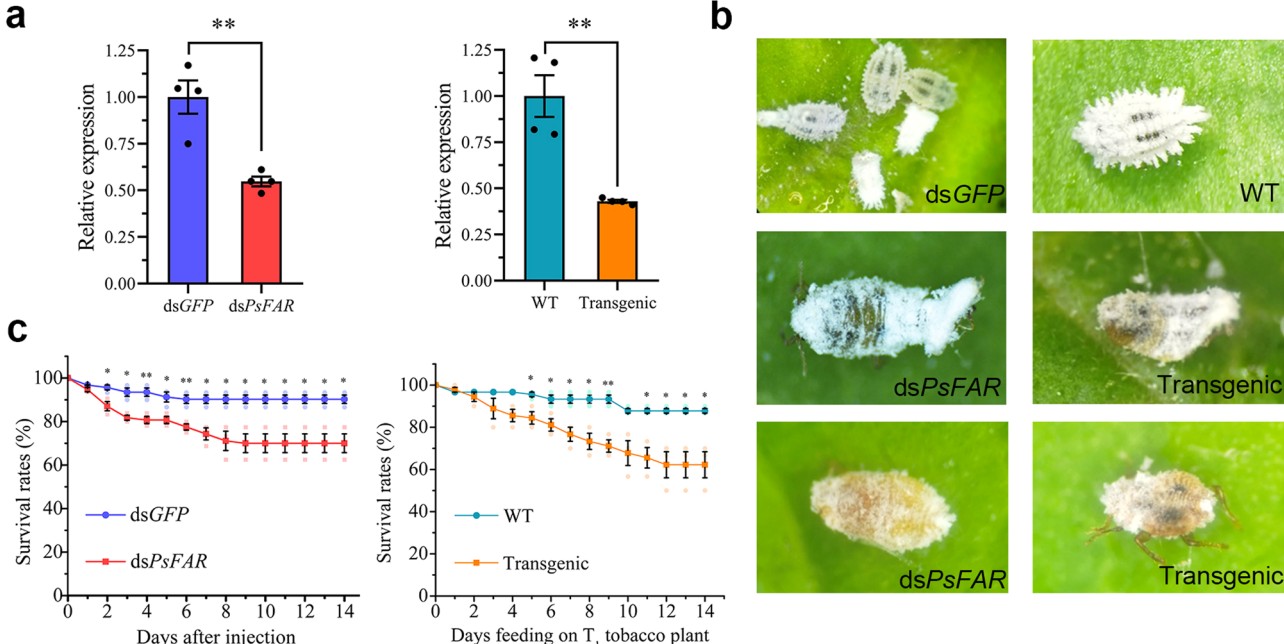

**Fig. 4 RNAi effects, lethality, and survival rate of *P. solenopsis* microinjected with dsRNA or fed on dsRNA expressing GM tobacco. a** Validation of RNAi effects of dsRNA microinjection (left) and GM tobacco feeding (right) using RT-qPCR. Total RNA for *PsFAR* was extracted from cotton mealybugs 3 days post injection ($n = 7$) and after feeding on GM tobacco for 5 days ($n = 10$). **b** Lethality in treated insects: ds*GFP* and WT tobacco-treated cotton mealybugs were normal and covered with a heavy waxy layer around their body surface, while ds*PsFAR* treated and GM tobacco treated insects exhibited abnormal coloration, reduced wax secretion, and unsuccessful molting. **c** Dynamic analysis of the survival rate of *P. solenopsis* after dsRNA injection (left) and feeding on tobacco plants (right). WT: insects fed on wild-type plants, Transgenic: insects fed on GM tobacco plants. ds*GFP* and WT were respectively set as negative controls, $n = 30$ insects. Means ± SEM were calculated with results from three biological replicates. Asterisks denote a significant difference as determined by Student's *t*-test; *$P < 0.05$, **$P < 0.01$.

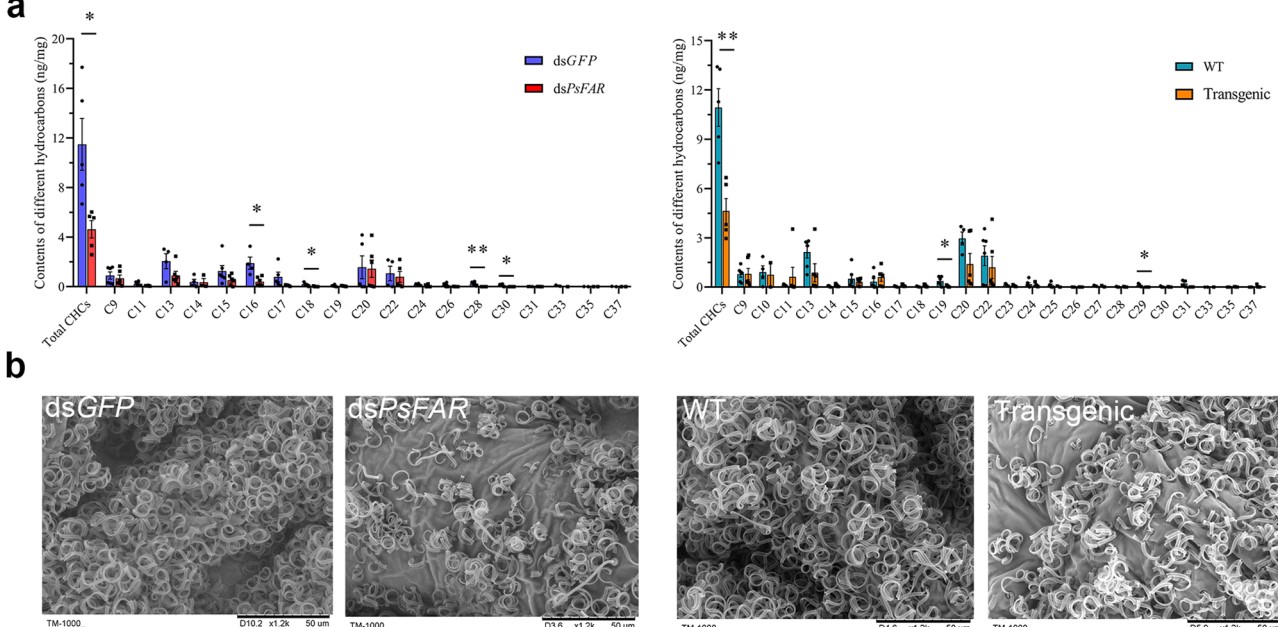

**Fig. 5 GC-MS and SEM analysis of wax. a** Effects of the relative CHC contents of *P. solenopsis* wax following dsRNA injection (left) and tobacco feeding (right). The contents of total CHCs and *n*-alkanes of different lengths ($C_9$–$C_{37}$) were analyzed by GC-MS. The *n*-heneicosane ($C_{21}$) was set as an internal standard. WT: insects fed on wild-type plants, Transgenic: insects fed on GM tobacco plants. Results are shown as nanograms per milligram of fresh body mass ± SEM from six biological replicates. * and ** respectively indicate significant differences at $P < 0.05$ and $P < 0.01$ between ds*GFP* and ds*PsFAR*, or WT and Transgenic (Student's *t*-test). **b** SEM analysis for dsRNA injection (left two) and tobacco rearing (right two) treated groups. 24 h post-emergence, live insects were collected for SEM. Images show the dorsal side of the cotton mealybug body from the third thoracic segment. Cotton mealybugs injected with ds*GFP* and fed on WT tobacco plants were used as negative controls.

numbers of *n*-alkanes were observed in both RNAi-treated groups (ds*PsFAR* and Transgenic) and the control groups (ds*GFP* and WT). As compared with the tobacco (WT and Transgenic) fed group, the microinjection (ds*GFP* and ds*PsFAR*) group lacked $C_{23}$, $C_{25}$, $C_{27}$, and $C_{29}$. In terms of CHC contents, the relative amounts of *n*-alkanes in RNAi-treated insects were significantly decreased, with the mean values of total *n*-alkanes per milligram of fresh body mass reduced by 6.86 and 6.3 ng/mg after ds*PsFAR* microinjection and GM tobacco feeding, respectively. This decrease likely resulted from reductions of different alkanes, particularly $C_{16}$, $C_{18}$, $C_{28}$, and $C_{30}$ in the ds*PsFAR* group and $C_{19}$ and $C_{29}$ in the Transgenic group, each of which decreased dramatically (Fig. 5a and Supplementary Table 4). These results reveal the critical involvement of *PsFAR* in CHC biosynthesis in the cotton mealybug.

As CHCs were found to be the dominant chemical component in cotton mealybug wax, we speculated that knocking down *PsFAR* might further affect wax output. To this end, 24-h-old female adults from the microinjection and GM tobacco-fed groups were subjected to scanning electron microscopy (SEM) and transmission electron microscopy (TEM) examinations. SEM observations from the third thoracic segment showed that all the ds*GFP*-microinjected and WT tobacco-fed insects were covered with a waxy layer on the epicuticular surface, but this waxy layer was reduced in both ds*PsFAR*-injected and GM tobacco-fed mealybugs (Fig. 5b). Additionally, the envelope integrity did not change evidently after knockdown of *PsFAR* as revealed by TEM analysis (Supplementary Fig. 6). Collectively, these results demonstrate that *PsFAR* has direct effects on wax biosynthesis in the cotton mealybug.

**Knockdown of *PsFAR* leads to reduced water retention**. To further investigate the functional relationships of *PsFAR* with water retention and waterproofing in cotton mealybugs, we carried out desiccation and water spray assays. The desiccation assay showed no significant difference in survival days for ds*PsFAR*-treated groups under 75% humidity compared with corresponding controls (Fig. 6a and Supplementary Table 5). In contrast, the survival days decreased dramatically in both ds*PsFAR* injected and GM tobacco-fed groups when insects were exposed to <10% humidity. However, for the water spray assay, the survival rate did not differ significantly between the controls and *PsFAR* knockdown groups (Supplementary Fig. 7). These results indicate that *PsFAR* participates in water retention but not waterproofing in the cotton mealybug.

**Knockdown of *PsFAR* increases the contact killing efficiency of insecticides**. We also investigated whether *PsFAR* is functionally important in preventing cotton mealybugs from contact-killing insecticides. As the concentration of deltamethrin decreased, the survival rate increased significantly in both female adults sampled from tomato plants and those from WT tobacco plants (Supplementary Fig. 8). At a concentration of 25 mg/L deltamethrin, the two groups had a survival rate of 78.89 ± 2.94% and 72.22 ± 2.94%, respectively, at 48 h after spraying with the insecticide. This concentration was therefore used in further contact-killing assays.

For groups not treated with deltamethrin, the survival rate decreased significantly in both ds*PsFAR* and GM tobacco-treated insects when compared to controls (Fig. 6b and Supplementary Table 5), suggesting that both groups would suffer increased mortality when *PsFAR* expression is suppressed. When treated with 25 mg/L deltamethrin, a more remarkable decrease in survival rate was observed 48 h after insecticide spray (Fig. 6b), which was 16.56% for ds*PsFAR* microinjected insects and 22.22%

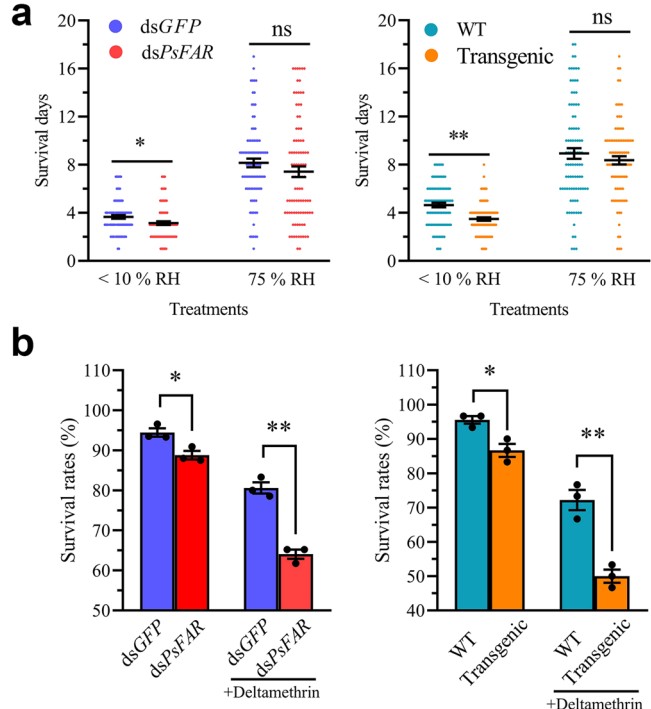

**Fig. 6 Effects of desiccation and insecticide treatment on the survival of cotton mealybugs. a** The survival rate of cotton mealybugs following dsRNA injection (left) and tobacco feeding (right) under conditions of 75% and <10% RH. Each circle represents one insect. $n = 30–34$ insects. **b** The survival rates of cotton mealybugs injected with dsRNA (left) and fed on tobacco plants (right). +Deltamethirin indicates insects in these groups were sprayed with 25 mg/L deltamethrin. $n = 30$ insects. WT: insects fed on wild-type plants, Transgenic: insects fed on GM tobacco plants. Data are presented as means ± SEM from three biological replicates. Significance was determined by the Student's *t*-test. *$P < 0.05$, **$P < 0.01$, ns: not significant.

for GM tobacco-fed insects. Our results suggest that *PsFAR* functions in protecting cotton mealybugs from insecticides.

## Discussion

As a group of highly aggressive and invasive insect pests, the waxy layer is crucial for mealybugs to adapt to differing environments. By focusing on the molecular mechanism responsible for wax biosynthesis, we demonstrated that CHCs are the dominant chemical components in the waxy layer of cotton mealybugs, and the biosynthetic pathways of these chemicals are associated with six DEGs that were significantly up-regulated in the integument. Particularly, we identified one female-enriched *FAR* gene, *PsFAR*, which plays a crucial role in the generation of CHCs and waxy filaments in the cotton mealybug. *PsFAR* was also found to contribute to the protective functions of the waxy layer, including water retention and as a barrier against contact insecticides.

Insect wax is a lipid-based substance that contains a wide variety of chemical compounds[32]. Here, the waxy filaments of cotton mealybug were determined to be a combination of hydrocarbons, fatty acids, olefins, alcohols, aromatic derivatives, esters, ketones, and aldehydes (Fig. 1b, Supplementary Fig. 1 and Supplementary Data 1). This was slightly different from that observed in the giant mealybug *Drosicha stebbingii* (Green) which contains hydrocarbons, esters, alcohols, ketones, and acids[10]. CHCs are the main components of cotton mealybug wax (Fig. 1b), as reported in many other insects[18,33]. As we used both polar (methanol) and nonpolar (hexane) solvents, as well as polar

and nonpolar chromatographic columns in GC–MS, the chemical compounds of the waxy filaments of cotton mealybug, should have been well characterized, according to Chen et al.[34]. Thus, the data reported here could prove very informative for future metabolomic analyses of the waxy filaments of mealybugs.

Among the CHC biosynthesis pathway-related DEGs, only *PsFAR* was highly expressed in third-instar and adult females (Fig. 2c and Supplementary Fig. 3), suggesting that this gene performs crucial functions in wax biosynthesis, as only females are known to produce this thick waxy barrier. The importance of *PsFAR* in wax biosynthesis can also be demonstrated by its expression sites. According to previous studies, oenocytes are the site for CHC biosynthesis in insects[18,35], and they exist in both the epidermis and fat body[36]; as such, genes responsible for wax biosynthesis would likely be highly expressed in these tissues. Consistently, in our study, the expression level of *PsFAR* in the fat body was significantly higher than in other tissues (Fig. 2c). It should be noted, however, that other *FAR*s identified in cotton mealybugs may play very different roles, as indicated by their expression profiles. For example, *PsFAR* I and *PsFAR* II, reported by Li et al.[28], are highly expressed in male adults (not covered by thick wax) and first- and second-instar nymphs, respectively. Such functional differences are supported by the phylogenetic tree provided in this study, where *PsFAR*, *PsFAR* I, and *PsFAR* II are clustered into three different branches (Fig. 2b).

To identify wax biosynthesis-related genes, we performed in vivo RNAi experiments to knock down *PsFAR* expression and observed reductions of CHCs and wax in RNAi-treated insects. This provided direct evidence not only for *PsFAR* involvement in wax biosynthesis but also for the significant contribution of CHCs content to the stability and function of mealybug wax. We paid careful attention to the potential effects of host plants on CHCs composition, as reported by Ahmad et al.[10]. Thus, we compared the difference in CHCs profiles between dsRNA microinjection groups that were reared on tomato or tobacco groups, where $C_{23}$, $C_{25}$, $C_{27}$, and $C_{29}$ were found to be absent in the former. These considerations made searching for *PsFAR* function more feasible and validated the credibility of our results.

In terms of *n*-alkane amounts, four *n*-alkanes ($C_{16}$, $C_{18}$, $C_{28}$, and $C_{30}$) in the ds*PsFAR* group and two *n*-alkanes ($C_{19}$ and $C_{29}$) in the transgenic group decreased significantly (Fig. 5 and Supplementary Table 4). The types of *n*-alkanes we identified were different from the findings in *N. lugens* injected with ds*NlFAR7* and ds*NlFAR9* (decreased *n*-alkanes were $C_{14-20}$, $C_{27}$, $C_{28}$, and $C_{29}$, and $C_{27}$ and $C_{29}$, respectively)[20,21]. Overall, only a small number of *n*-alkanes experienced changes in accumulation after the knockdown of *PsFAR* expression; a potential reason for this is that *PsFAR* might only regulate *n*-alkanes of a specific length. Meanwhile, decreased *n*-alkanes were those located in waxy filaments, but not those in the integumentary envelope because no obvious changes were detected in the envelope integrity of ds*PsFAR*-treated insects by TEM analysis (Supplementary Fig. 6). Yet, in *Drosophila*, *DmWP* is essential for morphogenesis of the tracheal envelope by regulating the formation of long-chain fatty alcohols which can serve as substrates for downstream CHCs biosynthesis[19]. Therefore, CHCs in the waxy filaments and envelope may be regulated by different *FAR* genes; in the cotton mealybug *PsFAR* mainly regulates CHCs content in the waxy filaments rather than the envelope. Functions of other *FAR* genes in the cotton mealybug remain unclear. According to the genome data, there are 22 annotated *FAR*s in the cotton mealybug[37]. Higher *FAR*s diversities have also been found in other insects. For example, there are 17 such genes in *N. lugens*[20], and 26 in *E. pela*[38]. Besides *FAR*s, other genes may also contribute to CHC biosyntheses, such as FAS[39,40], ELO[41,42], and cytochromes

P450[43–45]. Further research is needed to clarify the functions of these genes in cotton mealybugs and more deeply understand the biosynthetic mechanisms underlying cotton mealybug waxy filament production.

The waxy layer of the integument has been reported to reduce water loss through the cuticle[46–48] and similar results were observed in this study. When RNAi-treated mealybugs were kept under 75% humidity, their survival days did not differ significantly from the control groups, but decreased significantly at <10% humidity (Fig. 6a). This is different from the report in *N. lugens*, in which mortality due to low humidity was not affected by knockdown of *NlFAR*s[20,21]. Thus, as demonstrated, *FAR*s can have very diverse functions depending on insect species, and in cotton, mealybug plays a crucial role in wax biosynthesis. When *PsFAR* is knocked down, the mealybug suffers more water loss through the cuticle due to integument instability. One interesting finding is that some individuals can survive for more than 2 weeks without eating (Fig. 6a), reflecting an extremely strong adaptation of cotton mealybugs to severe environmental conditions[7,49].

After spraying with water, the survival rate of the *PsFAR* knockdown group did not change significantly as compared with the control groups (Supplementary Fig. 7), suggesting that *PsFAR* contributes little to waterproofing. This result is not strange, according to Blomquist and Ginzel[18], the lipids on the insect body surface have a high hydrophobicity. As such, even a small amount of wax on the cotton mealybug body surface could provide sufficient waterproofing against water spraying, despite the reduction in wax. This, again, supports our finding described above, that *PsFAR* does not influence the body envelope, as changes to the envelope would lead to higher mortality in the tested mealybugs.

RNAi-treated insects experienced a high mortality rate after spraying with a low concentration (25 mg/L) of deltamethrin (Fig. 6b; Supplementary Fig. 8), indicating that the waxy layer plays an important role in protecting cotton mealybugs from insecticide permeation. In other words, if the waxy filaments are reduced, such as by RNAi knockdown of *PsFAR* gene, the mortality of mealybugs would likely be in response to insecticide treatment. This further demonstrates the significance of *PsFAR* in wax biosynthesis in this mealybug.

RNAi has become a powerful tool for the study of gene functions, but off-target effects may occur in RNAi experiments[50]. In this study, to avoid off-target effects caused by dsRNA, two different dsRNA molecules targeting *PsFAR* were used in microinjection and oral delivery by transgenic plants, as suggested by Jarosch and Moritz[51]. Both dsRNAs showed satisfactory RNAi efficiency (Fig. 4a), similar to those obtained via other RNAi methods used in the cotton mealybug, such as siRNA microinjection[52] and direct dsRNA feeding[53]. Meanwhile, shared phenotypes produced by the two molecules further proved the same on-target effects for both of them. Additionally, the gene expression of *PsFAR* paralogs was not changed significantly before and after ds*PsFAR* treatment (Supplementary Fig. 5), and the two dsRNAs did not have matches when blasted against all other insect species, despite *PsFAR* clustering with other *FAR*s in the phylogenetic tree (Fig. 2b); thus it is almost impossible for the two dsRNA molecules having off-target effects on paralogous and orthologous genes of *PsFAR*. Together, these results indicate that off-target effects are less likely to occur in this study. To our knowledge, this is the first report to utilize a dsRNA-expressing transgenic host plant to induce RNAi in mealybugs and demonstrate that this technique is quite effective, as reported for other phloem-feeding hemipterans[54,55].

In summary, by performing multi-omics screening and functional validation, we identified an important *FAR* gene in the

cotton mealybug, *PsFAR*. Using two separate RNAi approaches, we showed that *PsFAR* plays a vital role in wax biosynthesis, which is essential for water retention and protects the insects from insecticide treatment. Our findings will hopefully deepen our understanding of the molecular mechanisms underlying wax biosynthesis. Further research is needed to identify key regulatory or interacting factors of FAR in mealybugs.

## Methods

**Insect rearing**. The cotton mealybugs used in this study were originally collected from Rose of Sharon, *Hibiscus syriacus* L. (Malvales: Malvaceae) in Jinhua, Zhejiang Province, China, in June 2016. They were maintained on fresh tomato plants (cv. Hezuo-903, Shanghai Changzhong Seeds Industry Co., Ltd, China) in a climatically controlled chamber maintained at 27 ± 1 °C, 75% relative humidity (RH), and a photoperiod of 14:10 (L:D). For detailed insect rearing and tomato cultivation methods see ref. [56].

**Scanning electron microscopy (SEM) of *P. solenopsis* wax**. SEM was used to observe changes in wax on the body surface of adult *P. solenopsis* females according to the methods of Huang et al.[57]. Briefly, collected insects were taped onto a stub and dried in an ion sputter (Hatachi, Tokyo, Japan) under a vacuum. After gold sputtering, the samples were observed using a TM-1000 SEM (Hatachi, Tokyo, Japan). Photos were scanned from the dorsal part of the third thoracic segment. Thirty insects were used for both RNAi-treated and control groups.

**Chemical composition analysis of mealybug wax**. A small soft brush was used to collect wax filaments from the body surface of *P. solenopsis* females. Prior to use, the brush was washed successively by 70% ethanol, sterile water, and 1× sterile phosphate-buffered saline (PBS, pH 7.4). The wax was collected into a clean chromatography vial for the following experiments. Two vials of wax, each collected from 1000 adult females, were dissolved in 1 ml of methanol and 1 ml of *n*-hexane, respectively. The vials were stirred gently for 3 min, kept at room temperature for 30 min, and then put into an S06H ultrasonic vibrator (Zealway, Xiamen, China) for 30 min to dissolve the wax sufficiently. The samples were analyzed on a TRACE 1310 (Thermo Scientific, Waltham, USA) gas chromatograph (GC) equipped with an ISQ single quadrupole MS and interfaced with the Chromeleon 7.2 data analysis system (Thermo Scientific, Waltham, USA), with a constant flow of helium at 1 ml/min. For each sample, a splitless injection of 1.0 μl was respectively made into a polar TG-WaxMS (Thermo Scientific, Waltham, USA) and a nonpolar TG-5MS (Thermo Scientific, Waltham, USA) 30 m × 0.25 mm × 0.25 μm capillary column. The temperature program for polar column samples was as follows: 40 °C for 2 min, then 5 °C/min to 240 °C, hold 10 min; the program for nonpolar column samples was: 40 °C for 2 min, then 5 °C/min to 300 °C, hold 5 min. Injector and detector temperatures were, respectively, set at 250 and 230 °C for polar column samples, and at 300 and 300 °C for nonpolar column samples. Mass detection for all samples was run under an EI mode with a 70 eV ionization potential and an effective *m/z* range of 35–450 at a scan rate of 5 scan/s. Chemical compounds were identified by mapping against the NIST database. The relative content of each compound was calculated by peak area which was determined using the Agilent MassHunter system.

**RNA extraction and RT-qPCR**. Total RNA was isolated using TRIzol reagent (Invitrogen, Carlsbad, CA) following the manufacturer's instructions, and RNA quality was accessed using agarose gel electrophoresis and a Biodrop μLite. 800 ng of total RNA was used for cDNA synthesis using the HiScript III RT SuperMixfor qPCR (+gDNA wiper) (Vazyme Biotech Co., Ltd., Nanjing, China), according to the manufacturer's instructions. Quantitative RT-PCR (RT-qPCR) was conducted using an AriaMx real-time PCR system (Agilent Technologies, USA), using a 20 μl reaction containing 2 μl of 10-fold diluted cDNA, 0.8 μl of each primer, and 10 μl ChamQ SYBR Color qPCR Master Mix (Vazyme Biotech Co., Ltd., Nanjing, China). The RT-qPCR thermocycling protocol was 95 °C for 30 s, followed by 40 cycles of 95 °C for 10 s and 60 °C for 30 s. The *PsActin* gene was used as an internal control. At least three biological replicates were used for each experiment. Quantitative variations were evaluated using the relative quantitative method ($2^{-\Delta\Delta Ct}$)[58].

**Transcriptome analysis of integumentary and non-integumentary tissues**. To obtain the integument and other tissues, adult *P. solenopsis* females were dissected in 1× sterile PBS (pH 7.4) on a sterile Petri dish. Dissected fresh tissues were directly used or frozen in liquid nitrogen and stored at −80 °C for follow-up experiments. We sequenced the transcriptomes of integumentary and non-integumentary tissues (all other tissues without integument) dissected from 150 adult females, with each sample being repeated in triplicate. mRNAs were purified from total RNA via oligo (dT) magnetic beads, and the fragmented mRNAs were then reverse transcribed into cDNA using random primers. Constructed pair-end libraries were sequenced using an Illumina HiSeq X Ten platform in Novogene (Beijing, China). After quality control, the clean RNA-Seq data of the six libraries were aligned with the *P. solenopsis* genome (http://v2.insect-genome.com/Organism/624) using HISTAT2[59]. Then featureCounts[60] and DESeq2[61] were used for the differential expression analysis of genes. The threshold for differentially expressed genes (DEGs) was defined by log2fold ≥ 1 or ≤−1 and a padj-value < 0.05.

**Proteome analysis of the integument**. Integuments dissected from 300 adult *P. solenopsis* females were ground in liquid nitrogen using a mortar and pestle, then dissolved in 400 μl SDT lysis buffer (4% SDS, 100 mM Tris–HCl, 1 mM DTT, pH 7.6). After sonication (ten pulses of 10 s with 10 s intervals, 100 W) and 15 min boiling, samples were centrifuged at 13,000×*g* at 4 °C for 40 min. Proteins were quantified by the BCA method (Solarbio, Beijing, China). Protein bands were checked by SDS–PAGE and Coomassie blue staining. In total, 300 μg of proteins were used in this experiment. The 100 mM DTT was removed by repeated ultra-filtration (10 kD microsep) using 200 μl of UA buffer (8 M urea, 150 mM Tris–HCl, pH 8.0), then 100 μl of iodoacetamide (100 mM in UA) was added to block reduced cysteine residues, and the samples were incubated for 30 min in the darkness. The filters were washed twice in 100 μl of UA buffer and twice in 100 μl of 25 mM $NH_4HCO_3$ buffer. Finally, the protein suspensions were digested with 6 μg trypsin (Promega) in 40 μl of 100 mM $NH_4HCO_3$ buffer for 16–18 h at 37 °C. The resulting peptides were desalted on C18 cartridges (Empore SPE Cartridges C18 (standard density), bed ID 7 mm, volume 3 ml) (Sigma-Aldrich, MO, USA), concentrated by vacuum centrifugation, and reconstituted in 40 μl of 0.1% (v/v) formic acid.

LC–MS/MS analysis was performed on a Q Exactive mass spectrometer (Thermo Fisher Scientific) coupled with Maxquant software (Thermo Fisher Scientific). Here, 2 μg of high pH reserved-phase peptide fragments were loaded onto a reverse-phase trap column (Thermo Fisher Scientific EASY column, 100 μm × 2 cm, 5 μm-C18) connected to the C18-reversed-phase analytical column (Thermo Fisher Scientific Easy Column, 75-μm inner diameter, 10-cm long, 3 μm resin) in buffer A (0.1% formic acid) and separated with a linear gradient of buffer B (0.1% formic acid and 84% acetonitrile) at a flow rate of 300 nl/min. The eluted peptides were ionized, and the full MS spectrum (from *m/z* 300 to 1800) was acquired by a precursor ion scan using the Q-Exactive analyzer with a resolution of $r = 70,000$ at *m/z* 200, followed by a 20 MS2 scan in the Q-Exactivea analyzer with a resolution of $r = 17,500$ at *m/z* 200. The MS raw files were translated into mgf files and searched against the integumentary transcriptome using Maxquant 1.3.0.5. Trypsin was defined as the cleavage enzyme allowing no more than two missed cleavages. Carbamidomethylation of cysteine was specified as a fixed modification, and oxidation of methionine was specified as a variable modification. Proteome extraction and sequencing were performed by Applied Protein Technology (Shanghai, China). To verify the existence of corresponding proteins expressed by integument upregulated DEGs, blast+ 2.12.0 was used to blast translated protein sequences of integumentary upregulated DEGs against the integumentary proteome data with the e-value set at 1E-5.

**Rapid amplification of cDNA ends of the *PsFAR* gene**. The SMART™ RACE cDNA Amplification Kit (Takara, Kyoto, Japan) was used to obtain full-length cDNAs of the *PsFAR* gene. 5′-UTR and 3′-UTR RACE cDNAs were synthesized from total RNA using SMARTScribe™ Reverse Transcriptase (Clontech), according to the manufacturer's instructions. Specific primers for *PsFAR* (Supplementary Table 6) were designed using Primer-BLAST in NCBI (https://www.ncbi.nlm.nih.gov/tools/primer-blast/). Paired with the Universal Primer Mix supplied in the kit, one pair of forward and reverse gene-specific primers were, respectively, used in the 3′ and 5′ RACE first-step PCR reactions. PCR conditions were as follows: incubation at 94 °C for 3 min; five cycles at 94 °C for 30 s, 72 °C for 3 min; five cycles at 94 °C for 30 s, 68 °C for 30 s, 72 °C for 3 min; and 25 cycles at 94 °C for 30 s, 66 °C for 30 s, 72 °C for 3 min. The final extension was 72 °C for 10 min. PCR products were purified using the FastPure Gel DNA Extraction Mini Kit (Vazyme Biotech Co., Ltd., Nanjing, China) and cloned using the pClone007 Simple Vector Kit (Tsingke Biotech, Beijing, China). Positive clones were selected and sequenced in Tsingke Biotech.

**Sequence analysis of *PsFAR***. The amino acid sequences of FARs were deduced from the corresponding cDNA sequence using ORFfinder in NCBI (https://www.ncbi.nlm.nih.gov/orffinder/). Multiple sequence alignments were conducted with ClustalX and GeneDoc. The phylogenetic tree was constructed in MEGA X[62], wherein neighbor-joining algorithm analysis using the JTT model for amino acids with 1500 bootstrap replicates was performed. Organisms and the GenBank accession numbers of sequences used here are shown in Supplementary Data 3.

**dsRNA synthesis and microinjection**. dsRNA was synthesized using the T7 High Yield RNA Transcription Kit (Vazyme Biotech Co., Ltd., Nanjing, China) according to the manufacturer's instructions. Briefly, the DNA template for dsRNA synthesis was amplified with primers containing the T7 RNA polymerase promoter at both 5′ ends (Supplementary Table 6). The purified DNA template (200 ng), a unique 435 bp fragment of *PsFAR*, was then used as templated for dsRNA production. ds*GFP* was used as a negative control. Synthesized dsRNAs were purified via isoamyl alcohol precipitation and re-suspended in nuclease-free water, and the concentration was quantified with a UV5NANO (Mettler-Toledo, Zurich,

Switzerland). Finally, the quality and size of dsRNAs were further verified via electrophoresis in a 1% agarose gel.

Microinjection of *P. solenopsis* was conducted with the Eppendorf InjectMan NI 2 microinjection system (Eppendorf, Hamburg, Germany). 4–5-day-old third-instar females were collected and pooled as a single biological replicate. After anaesthetizing with $CO_2$ for 30 s, ~200 ng of dsRNA was injected into the ventral thorax between the mesocoxa and hind coxa. After injection, insects were kept in a plastic rearing box (14 cm long, 10.5 cm wide, 5 cm high) containing one fresh tomato branch. Tomato branches were wrapped with moistened cotton at their base to provide water and renewed every 4 days.

**Vector construction and tobacco transformation of ds*PsFAR***. To express ds*PsFAR* in tobacco, the 422 bp fragment of *PsFAR* was first amplified using a forward primer containing *Bam*HI and *Sal*I restriction sites and a reverse primer containing *Sac*I and *Apa*I restriction sites (Supplementary Table 6). The sense and antisense strands of the purified *PsFAR* fragment were respectively double digested by *Sal*I + *Apa*I and *Sac*I + *Bam*HI. Two digested strands were inserted into corresponding restriction sites of the plant expression vector pCAMBIA1301[63] one at a time. The resulting RNAi transformation vector pCAMBIA1301-ds*PsFAR* was validated by sequencing (Tsingke Biotech, Beijing, China).

The pCAMBIA1301-ds*PsFAR* vector was introduced into tobacco (*Nicotiana tabacum* cv. Petit Havana) via *Agrobacterium*-mediated transformation. After transformation and culture, leaf discs were washed three times with distilled water and dried on absorbent paper. Then, leaf discs were pre-selected with kanamycin and transferred for differentiation and rooting. Regenerated plantlets were cultivated in a greenhouse for selection. Genetic transformation of tobacco was entrusted to Towin Biotechnology (Wuhan, China).

**Validation of ds*PsFAR* expression in GM tobacco plants**. The T5 Direct PCR Kit (Plant) (Tsingke Biotech, Beijing, China) was used to identify positive $T_0/T_1$ ds*PsFAR* genetically modified (GM) tobacco plants. Briefly, leaves 1–2 mm in diameter of $T_0/T_1$ GM and wild-type (WT) *N. tabacum* plants were lysed in 50 μl of lysis buffer A, followed by 10 min of incubation at 95 °C. After fully shocking by hand for 30 s and a brief centrifuge, the supernatant was used for PCR amplification. The 50 μl PCR reaction contained 1 μl of template DNA (i.e. supernatant), 2 μl of each primer and 25 μl of 2×T5 Direct PCR Mix (Plant) and the following thermocycling conditions were used: initial denaturation at 98 °C for 3 min; 35 cycles at 98 °C for 10 s, 63 °C for 10 s, 72 °C for 15 s; and final extension at 72 °C for 5 min. Primers used here are listed in Supplementary Table 6; the *RbcL* gene in vascular plants was used as a positive control. Genomic DNA from WT plants and double-distilled water were both used as negative controls. PCR products were analyzed by agarose gel electrophoresis.

To select for GM plants with the highest ds*PsFAR* expression for subsequent bioassays, total RNA of leaves from 9 randomly selected $T_0$ GM tobacco plants (50-day old) exhibiting similar growth were isolated. The relative expression levels of ds*PsFAR* were quantified by RT-qPCR, and the tobacco *EF-1A* gene was used as an internal control. All experiments were repeated in triplicate. $T_1$ GM plants (50-day-old) harvested from one $T_0$ GM plant that had the highest ds*PsFAR* expression were used for rearing newly-emerged third-instar nymphs. The cultivation process used for $T_1$ tobacco plants was the same as that used for tomato plants. Plants and mealybugs were reared under the same rearing conditions as described above.

**Assessing RNAi knockdown of *PsFAR* in *P. solenopsis***. RT-qPCR was performed to assess the effects of RNAi. For the ds*PsFAR* and ds*GFP* microinjection groups, seven *P. solenopsis* individuals were collected 3 days after injection from each group and used for RT-qPCR. For the groups reared on WT and GM tobacco plants, 10 third-instar nymphs already feeding on the plants for 5 days were collected for RT-qPCR analysis. Primers for *PsFAR* I and *PsFAR* II were cited from Li et al.[28]. Siblings were used for developmental observation and survival quantification. All experiments were performed in triplicate.

**Extraction and quantification of CHCs in *P. solenopsis* wax**. CHCs in *P. solenopsis* wax was extracted from adult females that were collected 4 days after injection, or after feeding on GM tobacco for 9 days, following a procedure modified from Li et al.[20,21]. Briefly, 50 females (approximately 15 mg) were placed in a clean chromatography vial and immersed in 200 μl *n*-hexane. After 3 min of sonication in an ultrasonic vibrator (S06H, Zealway, China), the solvent was drawn into a new clean chromatography vial. This procedure was repeated twice, and finally, 200 μl of hexane was used to rinse the nymphs and vial. All hexane extracts were combined, followed by 20 min of sonication. Undissolved impurities were pelleted by 10 min of centrifugation at 10,000 rpm, the supernatant was dried absolutely under high-purity nitrogen gas, then re-suspended in 30 μl of hexane. After adding 300 ng *n*-heneicosane ($C_{21}$) as an internal standard, samples were analyzed on the GC-MS system. The constant flow of helium was 1 ml/min. Splitless injection of 1.0 μl was made into a 30 m × 0.25 mm × 0.25 μm HP-5MS column (Agilent Technologies, Santa Clara, USA). The temperature program was operated as follows: 50 °C for 4 min, then 10 °C/min to 310 °C, hold 10 min. Injector and detector temperatures were respectively set at 270 °C and 300 °C. Mass detection was run under an EI mode with a 70 eV ionization potential and an

effective m/z range of 45–650 at a scan rate of 5 scan/s. A $C_7$–$C_{40}$ *n*-alkanes standard (Sigma-Aldrich, MO, USA) was analyzed using the same conditions. Chemical compounds were identified by mapping against the NIST database and calibrated against a standard. The peak area was determined by the Agilent MassHunter system. The relative content of each *n*-alkane was quantified using the following formula: peak area of *n*-alkane/peak area of *n*-heneicosane ($C_{21}$)*300. Six replicates for each treatment were performed.

**Transmission electron microscopy (TEM) of *P. solenopsis* integument**. To distinguish which pool of surface lipids was affected by *PsFAR*, we perform TEM to analyze the envelope integrity for adult females (24 h post-emergence, $n = 5$) sampled from both RNAi treated and control groups. The dissected integuments were first fixed in 2.5% glutaraldehyde overnight and rinsed three times with 0.1 M PBS (pH 7.0), 15 min for each time. After fixing with 1% osmium tetroxide for 1.5 h and rinsing twice with 0.1 M PBS (pH 7.0), samples were respectively dehydrated in an ethanol series (30 %, 50%, 70%, 80%, 90% and 95% (v/v)) for 15 min, followed by 20 min of dehydration in 100% ethanol and 100% acetone, respectively. Next, samples were impregnated (embedding agent: acetone 1:1, 1 h; embedding agent: acetone 3:1, 3 h; and pure embedding agent, 12 h), and embedded in Spurr resin (SPI-CHEM, USA) overnight at 70 °C. The ultrathin sections (70 nm) were prepared using an ultramicrotome (Leica, German). Following 0.1 M lead citrate staining, the samples were finally examined using TEM (JEM-2100plus, JEOL, Japan) operating at 200 kV of acceleration voltage.

**Desiccation assay**. To perform the desiccation tolerance bioassay, drying tubes with RH < 10% were prepared by putting approximately 10 g of packed fresh silica gel into a 50-ml centrifuge tube. In total, 30–34 adult females 4 days post injection or after feeding on GM tobacco for 9 days were put into a drying tube without food. In control experiments, females were starved at 75% RH in a normal 50-ml centrifuge tube. The humidity was assessed by a hygrometer (TH40W-EX, Miaoxin, Wenzhou, China). The rearing temperature was 27 ± 1 °C and the photoperiod was 14:10 (L:D). The number of surviving individuals was counted daily until all individuals died, and dead individuals were removed from the tube. Each treatment was performed in triplicate.

**Waterproofing assay**. For microinjected mealybugs, 30 adult females 4 days post-injection were collected and maintained in a rearing box. 500 μl of water was sprayed 10 cm straight down using a mini-sprayer (1.4 cm in diameter and 11.5 cm in height) (Supplementary Fig. 9) daily for 10 days. For mealybugs fed on GM tobacco plants, water was sprayed on plants directly using the same spray method. Both adaxial and abaxial surfaces of leaves harboring mealybugs were sprayed with water. The number of surviving *P. solenopsis* was recorded daily. The assay was repeated in triplicate.

**Insecticide assay**. To determine an appropriate concentration of insecticide for tolerance bioassays, groups of 30 one-day-old adult *P. solenopsis* females were collected from tomato or WT tobacco plants and placed in plastic boxes (same size as rearing box) that had been laid with a piece of filter paper at the bottom. For each box, 500 μl of an aqueous solution containing 25, 2.5 g/L, 250, 25, 2.5, and 0 mg/L of deltamethrin emulsion respectively was sprayed 10 cm straight down using the aforementioned mini-sprayer. One box with no treatment was set as the blank control. To reduce the effects of gastric toxicity, mealybugs were transferred into new rearing boxes immediately after spraying. The number of surviving mealybugs was counted three times within a 24 h interval. Each treatment was repeated in triplicate.

Based on the results obtained from the above assay, the 25 mg/L deltamethrin emulsion was used to evaluate the effects of *PsFAR* knockdown on insecticide resistance by *P. solenopsis*. The spray was performed on adult females 4 days post injection or after feeding on GM tobacco for 9 days following the same steps above. The surviving number of mealybugs was recorded 48 h after spraying. Assays for each treatment were performed in triplicate.

**Statistics and reproducibility**. All data sets were presented as mean ± SEM. Data were analyzed for statistical significance using analysis of variance (ANOVA) followed by Tukey's multiple comparison test (Figs. 2c, 3d, Supplementary Figs. 3 and 8), and using student's *t*-test (Figs. 4a, c, 5a, 6, Supplementary Figs. 5 and 7). Statistical significance is indicated with *p*-values as follows: *$P < 0.05$ and **$P < 0.01$. SPSS 19 and GraphPad Prism 8.0 were used for analyses.

**Reporting summary**. Further information on research design is available in the Nature Research Reporting Summary linked to this article.

## Data availability
The RNA-seq data of integumentary and non-integumentary tissues have been deposited in GenBank with the accession number PRJNA798788. The proteome data of the integument have been deposited in PeptideAtlas under the accession number

PASS01729. The sequence data of the pCAMBIA1301-dsPsFAR plasmid has been deposited in Addgene (#190937). The source data underlying graphs, plots, and charts in the manuscript are presented in Supplementary Data 4.

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

## Acknowledgements

This work was supported by the National Key R&D Program of China (2021YFD1400100), the National Natural Science Foundation of China (32102189, 31872029), and the China Postdoctoral Science Foundation (2020M681875). We thank Dr. Danting Li for her kind help in methodological guidance.

## Author contributions

F.L. and M.J. conceived, directed, and supervised the project. F.L., M.J. S.W. and H.T. designed the studies. S.D. collected samples for chemical composition analysis and proteomic sequencing. H.T., Zih.L. Zic. L., Y.A., and Yi.W. collected samples for RNA-seq. Zih.L. and H.T. analyzed the RNA-seq and proteomic data. Zih.L. constructed the recombinant expression vector for transgenic tobacco. H.T., Yu.W., M.A.A.O, and Zic.L. performed RNAi and bioassay experiments. H.T., F.L., and M.J. interpreted the data and wrote the manuscript. All authors have read and approved the manuscript submission.

## Competing interests

The authors declare no competing interests.
