## [Peer Review File · Communications Biology]

Reviewers' comments:

Reviewer #1 (Remarks to the Author):

In their manuscript entitled "The essential functions of the fatty acyl-CoA reductase gene in wax biosynthesis by the cotton mealybug, *Phenacoccus solenopsis* Tinsley" the authors Haojie Tong and colleagues report on the identification of a gene coding for fatty acid modifying enzyme (FAR) and the characterization of its organismal functions in the hemipteran pest *Phenacoccus solenopsis*. To evaluate the manuscript, I use the questions asked by the journal editors:

What are the major claims of the paper?

- The authors identify a FAR coding gene in the mealybug and show that it is to some extent essential for survival. They show that tobacco plants expressing dsRNA against FAR transcripts reduces the fitness of mealybug females. In particular, they become more sensitive to an insecticide and to desiccation. One should, however, contest the importance of FAR as a target for pest control as proposed in the abstract. Around 60% of mealybugs treated with dsRNA against FAR transcripts or fed on transgenic plants expressing dsRNA against FAR transcripts survive; the lethal effects are manifest after several days when insects have fed on the plant. To me, it seems that FAR is not a good candidate for pest control.

- The authors claim to have performed metabolomics: is this the CHC analyses? To me this is overrating of the CHC study.

Are they novel and will they be of interest to others in the community and the wider field?

- The data presented are not novel. The importance of FAR has been repeatedly shown in other insect species. The authors spend a lot of work on proteomics and transcriptomics to identify FAR as a key factor for cuticle wax production and function. No misunderstanding: I find this data very important; however, in this particular case, a simple look into the literature would have sufficed as an argument to work on FAR. Nevertheless, this work is certainly valuable to the community.

If the conclusions are not original, it would be helpful if you could provide relevant references. Is the work convincing, and if not, what further evidence would be required to strengthen the conclusions?

- This is the central critical point: the cuticle of some aphid species has two pools of surface lipids. One pool is constituted by the uppermost cuticle layer the envelope (TEM is desirable); the second pool harbours the curled wax filaments. The present work focusses only on the second pool; the first pool is completely neglected. Which pool is represented in the GC-MS assays? 70% of ethanol may also be effective in extracting envelope associated lipids. Indeed, alkanes etc are probably rather elements of the envelope. GC-MS after mechanical collection of the curled waxy filaments may serve to shed light on this matter. The barrier assays (water, insecticide) do not allow any conclusion on which pool is affected; the fewer curled wax filaments in FAR dsRNA treated insects may be due to a loss of envelope integrity (as an anchor for the curled waxy filaments) or a direct effect (as proposed by the authors). Together, the lipid provenance issue is not addressed properly, the work is incomplete.

- A second important point regards the sample collection for transcriptomics and proteomics: the integument in many insect species is associated with oenocytes that are lipid producing cells not only found close to the fat body. These cells were probably also present in the integument portion, not only in the fat body. On top of that, in many Aphids the curled waxy filaments are not deposited by the integument but specialised abdominal glands on the ventral side (if I remember well): The authors should at least discuss this issue. A better solution would be to provide in situ hybridisation data on FAR expression in tissues: is it expressed in the glands, the oenocytes, the integument?

- Finally, I find it noteworthy that males seem not to require FAR function. What is different in males? Do males also have the curled waxy filaments?

On a more subjective note, do you feel that the paper will influence thinking in the field?

- If the authors solve in detail the critical points just mentioned above, this work may contribute to our understanding of surface lipid function in insects.

Reviewer #2 (Remarks to the Author):

Using multi-omics analyses the authors investigate the role of a fatty acyl-CoA reductase (PsFAR) in cuticle maturation of mealybugs. The authors show that by knocking down the gene expression with RNAi the insects show higher mortality. Their evidence for that is based on rich a dataset that includes changes of CHC profiles, electron microscopy, toxicology assay, and desiccation assay using RNAi knockdown female insects. The authors have produced a compelling work that will be of interest to a broad audience studying developmental biology of insects, insect pest management and agriculture. Before acceptance, there are some issues that should be addressed as outline below.

- Abstract: at the end of the abstract the authors suggest that given the role of PsFAR for wax synthesis and, thus cuticle function, this gene may be a potential target for RNAi pest control. That is a strong sentence that should be carefully examined. RNAi is an important tool to investigate gene function in models where CRISPR is still not readily available. But RNAi is notorious for having off-targets. These off-targets could be paralogs or even orthologs in different insect species. Suggesting the use of RNAi to knockdown PsFAR with the intent of pest control should first involve a new study looking for off-targets in other insect species before it is made safe for pest control. Even insecticides that have been used for years and considered harmless to non-pest species were recently shown to impact beneficial insects (DOI: 10.7554/eLife.73812). The consequences to agriculture and environment could be catastrophic. I suggest removing this sentence from the abstract and, also suggest that authors discuss more the drawbacks of using RNAi in their manuscript.
- Supplementary material: I could not find a transcriptomics table showing the genes identified and read-count for every replicate/ condition. This material should be published along with the other supplementary material in case it hasn't yet. Supplementary table 2 is a summary only and not very informative.
- If different methods for GC-MS (hexane and methanol) were used the discussion should briefly comment on why is that, the drawbacks/ advantages of each one and, also the different results obtained.
- Line 241: Oenocytes are embedded in clusters in the epidermis or dispersed within the fat body cells (<https://doi.org/10.1038/s41437-020-00380-y>) – this may vary among insect species and between males and females of a given species. Lower expression of PsFAR in the integument does not rule out that epidermis could have an important role for the production and function of this enzyme. Please adjust the statement to include this information.
- Line 273: where is the reference for the statement that mealybugs can survive for more than two weeks without feeding?
- Line 289: RNAi techniques are well known for producing off-target effects. Here the authors used two different techniques to deliver the same dsRNA molecule. A better approach would be using two different dsRNA molecules targeting non-overlapping regions of PsFAR. If both molecules produced shared phenotypes that would assure that these phenotypes are produced by the same on-target. I would like to know what the authors comment on that is. I also suggest performing RT-qPCRs after dsRNA treatment (hopefully the authors still have the cDNA stored) for different genes. The authors could, for instance, check how the expression of PsFAR2 and a closer paralog (perhaps one of the genes from supplementary table 3) are affected by the PsFAR dsRNA previously used. If these genes show no difference in expression before and after dsRNA treatment that would not rule out the chance of other off-targets, but it would suggest off-targets are less likely to occur in the metabolic pathway investigated.
- Line 298: Still on the topic of RNAi it would be important to comment on the drawbacks of RNAi technique, and the studies that should be done before RNAi of PsFAR is proposed as pest control technique (look at the first comment about the abstract). Given the phylogenetic analysis that authors did it would be interesting to bring this data back to the discussion and comment on which species have closely related enzymes and what would be the chances of the dsRNA produced in this work find a match in any of these species.
- Line 316: How many insects were used for electron microscopy? Can the phenotypes be quantified and assessed by statistical analysis?
- Line 343: Include more information on RNA extraction and PCR. Was RNA quality assessed (gel? Nanodrop? Bioanalyzer?). Information on cDNA synthesis? Which kit was used and how much RNA used for cDNA synthesis? Include the reference for the $2(-\Delta\Delta Ct)$

(<https://doi.org/10.1038/nprot.2008.73>). Number and size of replicates?

- Line 500: was the temperature used for the desiccation assay also 27°C?

Point to point responses to reviewers

Responses to Reviewer #1

Reviewer #1 (Remarks to the Author):

In their manuscript entitled “The essential functions of the fatty acyl-CoA reductase gene in wax biosynthesis by the cotton mealybug, *Phenacoccus solenopsis* Tinsley“ the authors Haojie Tong and colleagues report on the identification of a gene coding for fatty acid modifying enzyme (FAR) and the characterization of its organismal functions in the hemipteran pest *Phenacoccus solenopsis*. To evaluate the manuscript, I use the questions asked by the journal editors:

Response: Thank you very much for your insightful and expert suggestions, which have significantly improved the quality of our manuscript. We have made corresponding modifications according to your suggestions.

What are the major claims of the paper?

- The authors identify a FAR coding gene in the mealybug and show that it is to some extent essential for survival. They show that tobacco plants expressing dsRNA against FAR transcripts reduces the fitness of mealybug females. In particular, they become more sensitive to an insecticide and to desiccation. One should, however, contest the importance of FAR as a target for pest control as proposed in the abstract. Around 60% of mealybugs treated with dsRNA against FAR transcripts or fed on transgenic plants expressing dsRNA against FAR transcripts survive; the lethal effects are manifest after several days when insects have fed on the plant. To me, it seems that FAR is not a good candidate for pest control.

Response: Many thanks for pointing out this problem. We have removed all sentences related to the issue of pest control in the revised manuscript.

- The authors claim to have performed metabolomics: is this the CHC analyses? To me this is overrating of the CHC study.

Response: No. The metabolomics we claimed here means the chemical components of the waxy filaments tested by GC-MS, see results in Fig. 1b, Supplementary Fig. 1 and Supplementary Data 1. We have adjusted the statements in the ‘Results’ and ‘Discussion’ sections to make it clear. (Lines 86-88, 223-234)

Are they novel and will they be of interest to others in the community and the wider field?

- The data presented are not novel. The importance of FAR has been repeatedly shown in other insect species. The authors spend a lot of work on proteomics and transcriptomics to identify FAR as a key factor for cuticle wax production and function. No misunderstanding: I find this data very important; however, in this particular case, a simple look into the literature would have sufficed as an argument to work on FAR. Nevertheless, this work is certainly valuable to the community.

Response: Though lots of work has been conducted to study FAR in insects, most of these reports only involve their functions in development and pheromone biosynthesis. Here, we provide direct evidence that *PsFAR* plays critical roles in wax biosynthesis in the cotton mealybug. This is the first report of a molecular mechanism of wax biosynthesis in mealybug species, a group of important invasive pests.

We have modified the statements in this revision to highlight the novelty, especially in the first three paragraphs of the 'Introduction'. As you mentioned, we believe this work is valuable to the community, and many thanks for your recognition.

If the conclusions are not original, it would be helpful if you could provide relevant references. Is the work convincing, and if not, what further evidence would be required to strengthen the conclusions?

- This is the central critical point: the cuticle of some aphid species has two pools of surface lipids. One pool is constituted by the uppermost cuticle layer the envelope (TEM is desirable); the second pool harbours the curled wax filaments. The present work focusses only on the second pool; the first pool is completely neglected. Which pool is represented in the GC-MS assays?

Response: Thanks for your keen insight into our data. We completely agree with your comments. The GC-MS for wax filaments represented only the second pool (Fig. 1b, Supplementary Fig. 1 and Supplementary Data 1), while the GC-MS for *n*-hexane extracted CHCs in Fig. 5 included two pools. We have now done the TEM analysis for the integument of cotton mealybug where the envelope was detected at the uppermost cuticle (see figures below and Supplementary Fig. 5).

Supplementary Figure 5 TEM analysis on the integumentary envelope. env: envelope, epi: epicuticle, pro: procuticle. (a) dsRNA injection treated group. (b) tobacco rearing treated group. WT: insects fed on wild type plants, Transgenic: insects fed on GM tobacco plants. 24 hours post emergence, live insects were collected for TEM.

70% of ethanol may also be effective in extracting envelope associated lipids. Indeed, alkanes etc are probably rather elements of the envelope. GC-MS after mechanical collection of the curled

waxy filaments may serve to shed light on this matter. The barrier assays (water, insecticide) do not allow any conclusion on which pool is affected; the fewer curled wax filaments in FAR dsRNA treated insects may be due to a loss of envelope integrity (as an anchor for the curled waxy filaments) or a direct effect (as proposed by the authors). Together, the lipid provenance issue is not addressed properly, the work is incomplete.

Response: Thank you for your suggestions. Our work mainly focusses on the second pool. GC-MS after mechanical collection of the curled waxy filaments is a great idea. However, because the second pool is a part of the cuticle and waxy filaments are also present in glands and pores, it is not possible to quantify and produce pollution-free extracts of CHCs from the second pool.

In the revised manuscript, we added the TEM experiments analyzing the envelope integrity as you suggested. Our results showed no obvious changes to the envelope after knockdown of *PsFAR* (see figures above and Supplementary Fig. 5), indicating few effects of *PsFAR* on the first pool, in other words, the decreased CHCs after knockdown of *PsFAR* mainly caused fewer curled wax filaments in the second pool.

As for the “70% of ethanol used for extracting envelope associated lipids”, it will be used in the future. Thanks.

- A second important point regards the sample collection for transcriptomics and proteomics: the integument in many insect species is associated with oenocytes that are lipid producing cells not only found close to the fat body. These cells were probably also present in the integument portion, not only in the fat body. On top of that, in many Aphids the curled waxy filaments are not deposited by the integument but specialised abdominal glands on the ventral side (if I remember well): The authors should at least discuss this issue. A better solution would be to provide in situ hybridisation data on FAR expression in tissues: is it expressed in the glands, the oenocytes, the integument?

Response: We completely agree with your point. Oenocytes are present in both the integument and fat body, we adjusted the statement to “According to previous studies, oenocytes are the site for CHC biosynthesis in insects (Blomquist and Ginzl, 2021; Makki et al., 2014), and they exist in both the epidermis and fat body (Holze et al., 2021); as such, genes responsible for wax biosynthesis would likely be highly expressed in these tissues. Consistently, in our study, the expression level of *PsFAR* in the fat body was significantly higher than in other tissues (Fig. 2c).” in the ‘Discussion’. (Lines 239-243)

Besides, the integument collected for transcriptomics and proteomics contains not only waxy filaments but also the glands inside. It is impossible to separate them by dissection, we have added this statement in the revision. (Lines 102-105)

- Finally, I find it noteworthy that males seem not to require FAR function. What is different in males? Do males also have the curled waxy filaments?

Response: Thanks for pointing this out. Males have no curled waxy filaments. We have added the sentence “Unlike adult males that have an elongated body with wings but no wax, female mealybugs are globose, flattened, wingless, and typically covered by a layer of thick, powdery wax.” in ‘Introduction’ to clarify this. (Lines 37-38)

On a more subjective note, do you feel that the paper will influence thinking in the field?

- If the authors solve in detail the critical points just mentioned above, this work may contribute to our understanding of surface lipid function in insects.

Response: Many thanks for your constructive comment. In the revision, we adjusted the statements to highlight the novelty and significance of our work and added TEM experiments to provide more evidence.

Responses to Reviewer #2

Reviewer #2 (Remarks to the Author):

Using multi-omics analyses the authors investigate the role of a fatty acyl-CoA reductase (PsFAR) in cuticle maturation of mealybugs. The authors show that by knocking down the gene expression with RNAi the insects show higher mortality. Their evidence for that is based on rich a dataset that includes changes of CHC profiles, electron microscopy, toxicology assay, and desiccation assay using RNAi knockdown female insects. The authors have produced a compelling work that will be of interest to a broad audience studying developmental biology of insects, insect pest management and agriculture. Before acceptance, there are some issues that should be addressed as outline below.

Response: We greatly appreciate your professional comments and giving us a chance to revise the manuscript.

- Abstract: at the end of the abstract the authors suggest that given the role of PsFAR for wax synthesis and, thus cuticle function, this gene may be a potential target for RNAi pest control. That is a strong sentence that should be carefully examined. RNAi is an important tool to investigate gene function in models were CRISPR is still not readily available. But RNAi is notorious for having off-targets. These off-targets could be paralogs or even orthologs in different insect species. Suggesting the use of RNAi to knockdown PsFAR with the intent of pest control should first involve a new study looking for off-targets in other insect species before it is made safe for pest control. Even insecticides that have been used for years and considered harmless to non-pest species were recently shown to impact beneficial insects (DOI: 10.7554/eLife.73812). The consequences to agriculture and environment could be catastrophic. I suggest removing this sentence from the abstract and, also suggest that authors discuss more the drawbacks of using RNAi in their manuscript.

Response: Thanks for your valuable comment. In the revised manuscript, we have removed all the sentences related to pest control and added one paragraph to discuss the off-target effects of RNAi in the 'Discussion', please see the penultimate paragraph.

- Supplementary material: I could not find a transcriptomics table showing the genes identified and read-count for every replicate/ condition. This material should be published along with the other supplementary material in case it hasn't yet. Supplementary table 2 is a summary only and not very informative.

Response: Added, please see Supplementary Data 2.

- If different methods for GC-MS (hexane and methanol) were used the discussion should briefly comment on why is that, the drawbacks/ advantages of each one and, also the different results obtained.

Response: Added, please see the second paragraph in the ‘Discussion’.

- Line 241: Oenocytes are embedded in clusters in the epidermis or dispersed within the fat body cells (<https://doi.org/10.1038/s41437-020-00380-y>) – this may vary among insect species and between males and females of a given species. Lower expression of PsFAR in the integument does not rule out that epidermis could have an important role for the production and function of this enzyme. Please adjust the statement to include this information.

Response: Thanks for pointing this out. We have adjusted the statement to “According to previous studies, oenocytes are the site for CHC biosynthesis in insects (Blomquist and Ginzl, 2021; Makki et al., 2014), and they exist in both the epidermis and fat body (Holze et al., 2021); as such, genes responsible for wax biosynthesis would likely be highly expressed in these tissues. Consistently, in our study, the expression level of *PsFAR* in the fat body was significantly higher than in other tissues (Fig. 2c).” and cited this reference. (Lines 239-243)

- Line 273: where is the reference for the statement that mealybugs can survive for more than two weeks without feeding?

Response: We apologize for missing the citation. It was based on the results of the desiccation assay (Fig. 6), we have added the citation “(Fig. 6)” after this sentence, but no references can be found for this. (Lines 290-292)

- Line 289: RNAi techniques are well known for producing off-target effects. Here the authors used two different techniques to deliver the same dsRNA molecule. A better approach would be using two different dsRNA molecules targeting non-overlapping regions of PsFAR. If both molecules produced shared phenotypes that would assure that these phenotypes are produced by the same on-target. I would like to know what the authors comment on that is. I also suggest performing RT-qPCRs after dsRNA treatment (hopefully the authors still have the cDNA stored) for different genes. The authors could, for instance, check how the expression of PsFAR2 and a closer paralog (perhaps one of the genes from supplementary table 3) are affected by the PsFAR dsRNA previously used. If these genes show no difference in expression before and after dsRNA treatment that would not rule out the chance of other off-targets, but it would suggest off-targets are less likely to occur in the metabolic pathway investigated.

Response: We completely agree with your point about RNAi off-target effects. We have discussed this issue in the penultimate paragraph of the ‘Discussion’. Actually, we used two different dsRNA molecules, please see the primers listed in Supplementary Table 6. We have corrected the dsRNA length in the revised manuscript. (Line 453)

Regarding the gene expression of *FAR* paralogs, we have added experiments to check the expression of *PsFAR1*, *PsFAR2* (reported by Li et al., 2016) and another *FAR* gene after knockdown of *PsFAR*, no significant changes of gene expression were detected for these genes (please see figures below and Supplementary Figure 4), indicating the two dsRNA molecules had little effect on paralogous genes of *PsFAR*.

Supplementary Figure 4 Gene expression of *FAR* paralogs in cotton mealybug. (a) *PsFAR* identified in this study. (b) and (c) represent *PsFAR1* and *PsFAR2* identified by Li et al., (2016). The relative expression level was normalized and visualized as the means \pm SEM of three biological replicates. WT: insects fed on wild type plants, Transgenic: insects fed on GM tobacco plants. ns denote no significant difference as determined by Student's *t*-test with $P=0.05$.

- Line 298: Still on the topic of RNAi it would be important to comment on the drawbacks of RNAi technique, and the studies that should be done before RNAi of PsFAR is proposed as pest control technique (look at the first comment about the abstract).

Response: Thanks for your suggestions. We have commented on the drawbacks of the RNAi technique (please see the penultimate paragraph of Discussion) and deleted all the sentences related to pest control.

Given the phylogenetic analysis that authors did it would be interesting to bring this data back to the discussion and comment on which species have closely related enzymes and what would be the chances of the dsRNA produced in this work find a match in any of these species.

Response: Thanks. No matches of the two dsRNAs were found in any other species from NCBI. We have added this sentence in the 'Discussion' section. (Lines 315-317)

- Line 316: How many insects were used for electron microscopy? Can the phenotypes be quantified and assessed by statistical analysis?

Response: Our apologies. We have added the sample size for SEM in the revision as this: "Thirty insects were used for both RNAi-treated and control groups." (Lines 343-344) The observed wax covering the body surface can only be qualitative by electron microscopy.

- Line 343: Include more information on RNA extraction and PCR. Was RNA quality assessed (gel? Nanodrop? Bioanalyzer?). Information on cDNA synthesis? Which kit was used and how much RNA used for cDNA synthesis? Include the reference for the 2(-ddCt) (<https://doi.org/10.1038/nprot.2008.73>). Number and size of replicates?

Response: RNA quality was assessed by gel electrophoresis and Nanodrop. 800 ng of total RNA was used for cDNA synthesis according to the manufacturer's instructions for the HiScript III RT SuperMix for qPCR (+gDNA wiper) (Vazyme Biotech Co., Ltd., Nanjing, China). At least three

biological replicates were included for each experiment. All this information has been added to the 'Materials and Methods' section. (Lines 370-379)

- Line 500: was the temperature used for the desiccation assay also 27°C?

Response: Yes, we have changed this sentence to "Rearing temperature was 27 ± 1 °C and photoperiod was 14:10 (L:D)" to make it clear. (Line 548)

REVIEWERS' COMMENTS:

Reviewer #1 (Remarks to the Author):

Thank you to the authors for addressing my points of critics. However, I would like to ask the authors to provide low-magnification TEM-images shown in Supplementary figure 5. The pore/wax canals seem to be interesting and would probably would require an interpretation.

Reviewer #2 (Remarks to the Author):

The manuscript by Tong et al. has been clearly improved in this revised version and the authors carefully addressed all my recommendations. This manuscript contains a rich dataset that will be of interest to a broad audience. My main concern was the possibility of RNAi off-targets. This has been addressed by the clarification that two different dsRNA molecules were used, and new data showing that no effect on the expressions of paralogs (PsFarI and PsFarII) is observed. That minimizes the chances of off-targets and the authors included a important paragraph to the discussion about the topic. The authors also removed from the manuscript their hypothesis that PsFar could be a target to pest control. I agree with the their decision and consider it a sensible one. Removing it does not diminish the importance of their work. I am satisfied with this revised version.

Point to point responses to reviewers

Reviewer #1 (Remarks to the Author):

Thank you to the authors for addressing my points of critics.

However, I would like to ask the authors to provide low-magnification TEM-images shown in Supplementary figure 5. The pore/wax canals seem to be interesting and would probably would require an interpretation.

Response: Thank you very much for your suggestions. We have added the low-magnification (2 μ m/1 μ m and 500 nm) TEM-images in Supplementary figure 5. Please see figures below and Supplementary figure 5. Since the pore/wax canals are not the main topic here, we wish to leave the more discussion to next research. Thanks very much.

Supplementary Figure 5 TEM analysis on the integumentary envelope. env: envelope, epi: epicuticle, pro: procuticle. (a) *dsGFP* injection treated group. (b) *dsPsFAR* injection treated group. (c) WT tobacco rearing treated group. (d) Transgenic tobacco rearing treated group. WT: insects fed on wild type plants, Transgenic: insects fed on GM tobacco plants. 24 hours post emergence, live insects were collected for TEM.

Reviewer #2 (Remarks to the Author):

The manuscript by Tong et al. has been clearly improved in this revised version and the authors carefully addressed all my recommendations. This manuscript contains a rich dataset that will be of interest to a broad audience. My main concern was the possibility of RNAi off-targets. This has been addressed by the clarification that two different dsRNA molecules were used, and new data showing that no effect on the expressions of paralogs (PsFarI and PsFarII) is observed. That minimizes the chances of off-targets and the authors included a important paragraph to the discussion about the topic. The authors also removed from the manuscript their hypothesis that PsFar could be a target to pest control. I agree with the their decision and consider it a sensible one. Removing it does not diminish the importance of their work. I am satisfied with this revised version.

Response: Great thanks for your positive comments. Thanks for your time and expertise in reviewing our work.